# Mechanism and Kinetic Analysis of Degradation of Atrazine by US/PMS

**DOI:** 10.3390/ijerph16101781

**Published:** 2019-05-20

**Authors:** Yixin Lu, Wenlai Xu, Haisong Nie, Ying Zhang, Na Deng, Jianqiang Zhang

**Affiliations:** 1College of Architectural and Environmental Engineering, Chengdu Technological University, Chengdu 611730, China; yxlu61@163.com (Y.L.); txgsfy@163.com (Y.Z.); nstxdy@163.com (N.D.); 2Center of Big Data for Smart Environmental Protection, Chengdu Technological University, Chengdu 611730, China; 3Faculty of Geosciences and Environmental Engineering, Southwest Jiaotong University, Chengdu 610031, China; zhangchengdu@126.com; 4State Key Laboratory of Geohazard Prevention and Geoenvironment Protection, Chengdu University of Technology, Chengdu 611730, China; 5Haitian Water Grp Co Ltd., Chengdu 610059, China; 6Department of International Environmental and Agricultural Science, Tokyo University of Agriculture and Technology, Tokyo 1838509, Japan; nie-hs@cc.tuat.ac.jp

**Keywords:** ultrasound, peroxymonosulfate, free radicals, ATZ, degradation mechanism

## Abstract

The degradation effect, degradation mechanism, oxidation kinetics, and degradation products of Atrazine (ATZ) by Ultrasound/Peroxymonosulfate (US/PMS) in phosphate buffer (PB) under different conditions were studied. It turned out that the degradation rate of US/PMS to ATZ was 45.85% when the temperature of the reaction system, concentration of PMS, concentration of ATZ, ultrasonic intensity, and reaction time were 20 °C, 200 μmol/L, 1.25 μmol/L, 0.88 W/mL, and 60 min, respectively. Mechanism analysis showed that PB alone had no degradation effect on ATZ while PMS alone had extremely weak degradation effect on ATZ. HO• and SO_4_^−^• coexist in the US/PMS system, and the degradation of ATZ at pH7 is dominated by free radical degradation. Inorganic anion experiments revealed that Cl^−^, HCO_3_^−^, and NO_3_^−^ showed inhibitory effects on the degradation of ATZ by US/PMS, with Cl^−^ contributing the strongest inhibitory effect while NO_3_^−^ showed the weakest suppression effect. According to the kinetic analysis, the degradation kinetics of ATZ by US/PMS was in line with the quasi-first-order reaction kinetics. ETA with concentration of 1 mmol/L reduced the degradation rate of ATZ by US/PMS to 10.91%. Product analysis indicated that the degradation of ATZ by US/PMS was mainly achieved by dealkylation, dichlorination, and hydroxylation, but the triazine ring was not degraded. A total of 10 kinds of ATZ degradation intermediates were found in this experiment.

## 1. Introduction

Atrazine (ATZ) is one of the most widely used chemical herbicides in Asian and South American countries. The domestic use amount of ATZ was about 1.8 × 10^6^ kg in late 1990s and increased by an average of 20% each year [1]. At this level, the annual use amount of ATZ in China could reach 10^8^ kg by the end of 2018. ATZ can transform through different environmental media by diffuse, volatile, surface runoff, leaching, dry and wet deposition, etc. The ways ATZ entering water mainly include surface runoff, leaching, and dry and wet deposition [2]. The structure of ATZ is stable and difficult to degrade in the natural environment, and it is also hard to mineralize by microorganisms [3]. Its half-life in surface water can be up to 700 days [4]. ATZ with concentration up to 108 μg/L has been detected in north American rivers [5]. ATZ with a concentration of 3.9–81.3 μg/L has been detected in many rivers and reservoirs in China [6,7,8,9,10], which far exceeded the limit of 2 μg/L according to the hygienic standard of drinking water in China.

The main effects of ATZ on biology are toxic effects and endocrine interference. According to the research by Benjamin [11], ATZ can cause stunted vertebrae development of zebra fish, and excessive ATZ (more than 3 mmol/L) can cause serious defects in the craniofacial development of zebrafish. The research conducted by HAYES [12] and others found that, when the growth of the African tree frog from larva to adult are exposed to the ATZ solution with the concentration of 0.1 μg/L, 20% of the young frogs will develop into hermaphrodites, which indicates that ATZ with a low concentration can feminize the male African tree frog. The team of Jia [13] indicated that ATZ can further cause heart and liver damage in quails by causing ion disturbances. Meanwhile, the study by Sanderson [14] and others discovered that ATZ can interfere with endocrine balance by increasing the activity of the CYP19 enzyme in the human body.

Environmental atrazine removal mainly involves several kinds of methods including biodegradation, chemical degradation, and physical absorption. Ma et al. [15] reported that an *Ensifer* sp. strain can mineralize atrazine completely from the surrounding environment by utilizing it as its main nitrogen source. He et al. [16] reported the effects of fulvic acids and electrolytes on colloidal stability and photocatalysis of nano-TiO_2_ for atrazine removal. The results indicate that the removal efficiency of atrazine by nano-TiO2 decreased with the increase in fulvic acids concentration. Wu et al. [17] investigated the removal efficiency of atrazine from aqueous solutions using magnetic Saccharomyces cerevisiae bio-nanomaterial, and found that the maximum removal efficiency of 100% was achieved at 28 °C, a pH of 7.0, and 150 rpm with an initial atrazine concentration of 2.0 mg/L. In another research, Zhu et al. [18] pointed out that both adsorption and biodegradation by the bio-nanocomposite contributed to atrazine removal. Zhao et al. [19] studied the sorption properties of biochars (CS450 and ADPCS450) from corn straw with different pretreatment and sorption behavior of atrazine. The sorption experiment showed that sorption was more favorable for atrazine sorption in acidic solution and the sorption was temperature-dependent and a spontaneous process. At present, the photo-Fenton and photo-Fenton-like advanced oxidation technologies based on Persulfate/Peroxymonosulfate (PS/PMS) have been proven to be effective in the degradation of ATZ in water including Heat/PS [20], PBS/PMS [21], UV/PS [22], and UV/PMS [23]. For example, Ji et al. [20] demonstrated that heat-activated persulfate could effectively degrade ATZ in water and pointed out that the increase of the persulfate concentration or temperature significantly enhanced the degradation efficiency. Khan et al. investigated the efficacy of atrazine degradation by UV, UV/H_2_O_2_, UV/PS, and UV/PMS. However, so far, there have been few reports on the degradation of ATZ by US/PMS. Therefore, in this paper, the effect of US/PMS oxidative degradation to ATZ under different conditions was investigated in PB solution, and its degradation mechanism, oxidation kinetics, and degradation products were analyzed and discussed. These factors have a certain reference value for chemical treatment of pesticide wastewater.

## 2. Materials and Methods

### 2.1. Reagents and Instruments

Reagents: methyl alcohol of chromatographically pure, sodium hydroxide, sodium dihydrogen phosphate, sodium nitrite, ethanol, tert-butanol, sodium chloride, sodium bicarbonate, potassium nitrate, which are all analytically pure. ATZ and PMS (KHSO_5_·0.5KHSO_4_·0.5K_2_SO_4_, KHSO_5_ ≥ 47%) were purchased from Aladdin Co., Ltd. (Seoul, Korea).

Instruments: high performance liquid chromatograph (Waters 2695-2996), electronic scales, ultrasonic generator with a frequency of 150 KHz, lab pH meter from Shanghai Electronics Science Instrument Co., Ltd., Shang Hai, China, NC ultrasonic cleaner (KH5200DB), Ultra-pure Water Purifier from ULUPURE Co., Ltd., Chengdu, China, energy-saving intelligent thermostat tank (DC-1030), stirring hot plate (78 HW-1).

### 2.2. Experimental Scheme

#### 2.2.1. Solution Preparation

Ultrapure water with the electrical resistivity of 18.24 MΩ·cm was used to prepare ATZ mother liquor of 100 μmol/L, NaH_2_PO_4_ solution of 0.2 mol/L, NaOH solution of 0.02 mol/L, NaNO_2_ solution of 0.1 mol/L, NaCl solution of 1 mol/L, NaHCO_3_ solution of 0.5 mol/L, KNO_3_ solution of 1mol/L, PMS solution of 0.01 mol/L in sealed containers out of light, tertiary butanol solution of 16 g/L, and ethanol solution of 16 g/L. The preparation methods of phosphate buffer at pH 6, pH 7, and pH 8 with constant volume of 1L were shown in Table 1.

#### 2.2.2. Experimental Scheme of ATZ Degradation by US/PMS

The degradation efficiency of US/PMS to ATZ in 1.25 mmol/L phosphate buffered solution was investigated under the following conditions: different temperature of 10, 15, 20, and 25 °C, different pH value of 6, 7, and 8, different PMS concentration of 50, 100, 200, and 400 μmol/L, different ATZ concentration of 0.625, 1.25, and 2.5 μmol/L, and different US intensity with the frequency of 150 KHz of 0.22 W/mL, 0.44 W/mL, 0.66 W/mL, and 0.88 W/mL.

Different concentrations of tertiary butanol and ethanol were added to investigate the mechanism of degradation of ATZ by US/PMS in 20 °C water bath when the US intensity, PB concentration, PMS concentration, and ATZ concentration were 0.88 W/mL, 1.25 mmol/L, 200 μmol/L, and 1.25 μmol/L, respectively. The effect of common anions in water including Cl^−^, HCO_3_^−^, and NO_3_^−^ on US/PMS degradation to ATZ was investigated by adding different concentrations of NaCl, NaHCO_3_, and NaNO_3_ solution. NaNO_2_ solution of 0.1 mol/L served as the termination agent for the reaction.

### 2.3. Analytical Method

Symmetry^®^ C18 stable bond was adopted to detect ATZ, and the specific test method is as follows: the mobile phase ratio of methyl alcohol to ultrapure water is 60:40, flow velocity of 0.8 mL/min, column temperature of 40 °C, and wavelength of 225 nm.

## 3. Results and Discussion

### 3.1. Effect of Temperature on ATZ Degradation by US/PMS

The influence of temperature on ATZ degradation by US/PMS in phosphate buffered solution at pH 7 was shown in Figure 1 when the concentration of ATZ, US intensity, and PMS concentration were respectively 1.25 μmol/L, 0.88 W/mL, and 200 μmol/L. In this figure and all the following figures, *C* represents ATZ concentration at any time and *C*_0_ is the concentration of ATZ at time 0. According to Figure 1, with the increase of reaction temperature, the effect of ATZ degradation by US/PMS was enhanced. The ATZ removal rate increased from 19.37% to 50.96% when the reaction temperature rose from 10 °C to 25 °C. This is mainly due to the increase in the percentage of PMS molecules activated by a temperature rise, which accelerates the decomposition of PMS to generate SO_4_^−^• and HO• [24]. At the same time, increasing the temperature will accelerate the motion speed of molecules and increasing the collision frequency between ATZ, SO_4_^−^•, and HO•, accelerates the degradation of ATZ. It is easy to observe that, when the temperature increased from 15 °C to 20 °C, the removal rate of ATZ increased more significantly. When compared with the temperature from 10 °C to 15 °C and from 20 °C to 25 °C, the improvement effect clearly indicates that the temperature change within the normal range has a great impact on US/PMS degradation to ATZ. As the reaction temperature increases, the effect of temperature variation on the US/PMS based removal rate of ATZ will be less pronounced. Su et al. also observed the same phenomenon [25] in the research of the degradation efficiency of the antibiotics amoxicillin in aqueous solution. They found that the activation of sulphate radicals under ultrasound irradiation present nonlinear correlation with a temperature variation.

### 3.2. The Effect of PMS Concentration on the ATZ Degradation by US/PMS

The influence of PMS concentration on the ATZ degradation by US/PMS was demonstrated in Figure 2 when the ATZ concentration, US intensity, and temperature were 1.25 μmol/L, 0.88W/mL, and 20 °C. As shown in Figure 2, the effect of ATZ degradation by US/PMS was enhanced with the increase of PMS concentration. The ATZ removal rate increased from 28.90% to 58.77% when the PMS concentration rose from 50 μmol/L to 400 μmol/L. It is vital because increasing PMS concentration in the reaction system will relatively increase the yield of SO_4_^−^• and HO• per unit time, and then accelerate ATZ degradation by US/PMS when other conditions remain unchanged. It is worth mentioning that, when the concentration of PMS increased from 100 μmol/L to 200 μmol/L, the removal rate of ATZ did not change much. Therefore, PMS concentration has a greater influence on US/PMS degradation to ATZ.

### 3.3. The Effect of pH Value on the ATZ Degradation by US/PMS

The impact results of a different pH value on the ATZ degradation by US/PMS in phosphate buffer solution at 20 °C were shown in Figure 3 when the concentration of ATZ. US intensity and PMS concentration were 1.25 μmol/L, 0.88 W/mL, and 200 μmol/L. From Figure 3, the effect of ATZ degradation by US/PMS was enhanced with the increase of pH value. The ATZ removal rate increased from 46.46% to 56.77% when the pH value rose from 6 to 8. The ATZ removal rate was higher in alkaline pH than in acidic pH, primarily because ultrasound can stimulate PMS to generate SO_4_^−^• and HO• simultaneously (Equation (1) shows the reaction equation), and HO• has a slightly stronger oxidation capacity to ATZ than SO_4_^−^•. The secondary reaction rates of the two with ATZ were 3 × 10^9^ M^−1^·s^−1^ [26] and 2.59 × 10^9^ M^−1^·s^−1^ [23], respectively. In aqueous solution, SO_4_^−^• can react with water at any given pH condition to produce HO•, and the reaction rate constant is 8.30 M^−1^·s^−1^ [27] (see Equation (2) for the reaction equation). Under alkaline conditions, SO_4_^−^• can also react with OH^−^ to produce HO•, and the reaction rate constant is 6.50 × 10^7^ [28] (The equation is shown in Equation (3)), and the change of pH did not affect the yield of the SO_4_^−^• in US/PMS. Thus, under alkaline conditions, the US/PMS system produced more HO• per unit time. Therefore, the removal rate of ATZ is higher in alkaline pH than in acidic conditions. Thus, it can be seen that pH has a greater influence on ATZ degradation by US/PMS [23,26,28].
HSO_5_^−^+US→SO_4_^−^•+HO• *K*_HO•_=3×10^9^ M^−1^·s^−1^; *K*_SO4_^−^_•_ = 2.59 × 10^9^ M^−1^·s^−1^(1)
SO_4_^−^•+H_2_O→SO_4_^2-^+HO•+H^+^*K* = 8.30 M^−1^·s^−1^(2)
SO_4_^−^•+OH^−^→SO_4_^2-^+HO• *K* = 6.50 × 10^7^(3)

### 3.4. The Effect of US Intensity on the ATZ Degradation by US/PMS

The influence of different US intensity on the ATZ degradation by US/PMS in PB at 20 °C were shown in Figure 4 when the ATZ concentration, PMS concentration, and pH were, respectively, 1.25 μmol/L, 200 μmol/L, and 7 μmol/L. As seen in Figure 4, the effect of US/PMS to ATZ degradation gradually increased with the increase of US intensity in the reaction system. The removal rate of ATZ increased from 25.53% to 45.85% when the US intensity increased from 0.22 W/mL to 0.88 W/mL. This is mainly due to the fact that the ultrasound can cause a cavitation bubble phenomenon, while the formation and collapse of the cavitation bubble can form extreme high temperature and pressure conditions and then activate PMS to generate SO_4_^−^• and HO• to degrade ATZ [25,29]. It is worth noting that, when the US intensity increased from 0.66W/mL to 0.88 W/mL, the removal rate of ATZ was significantly higher than when the US intensity increased from 0.22 W/mL to 0.44 W/mL. The results showed that US intensity had a great effect on US/PMS degradation to ATZ.

### 3.5. The Effect of ATZ Concentration on the ATZ Degradation by US/PMS

The influence of different ATZ intensity on the ATZ degradation by US/PMS in PB at 20 μmol/L were shown in Figure 5 when the pH, US intensity, and PMS concentration were, respectively, 7, 1.25 μmol/L, 0.88 W/mL, and 100 μmol/L. From Figure 5, with the increase of ATZ concentration in the reaction system, the effect of US/PMS degradation ATZ was gradually weakened. The ATZ removal rate decreased from 71.37% to 35.22% when the ATZ concentration increased from 0.625 μmol/L to 2.5 μmol/L. It is easy to observe that, when the ATZ concentration increased from 0.625 μmol/L to 1.25 μmol/L, the ATZ removal rate decreased significantly from 71.37% to 40.88%. The ATZ removal rate decreased slightly from 40.88% to 35.22% when the ATZ concentration increased from 1.25 μmol/L to 2.5 μmol/L. The results showed that ATZ concentration had a significant effect on ATZ degradation by US/PMS.

### 3.6. Mechanism Analysis of ATZ Degradation by US/PMS

The mechanism of ATZ degradation by US/PMS was analyzed by using a single factor method, and the results are shown in Figure 6 where the concentration of PB in pH7, US intensity, PMS concentration, temperature, and ATZ concentration were 1.25 mmol/L, 0.88 W/mL, 200 μmol/L, 20 °C, and 1.25 μmol/L. According to Figure 6a, PB alone had no degradation effect on ATZ, but PMS alone had a very weak degradation effect on ATZ at the current concentration, and the degradation rate was 4.1%. The degradation efficiency of ATZ by US/PB/PMS (The US/PB/PMS system and US/PMS system are distinguished in this section, and the US/PMS system in other places refers to PB) alone was 13.43%, which accounts for 29.29% of the total removal rate of US degradation ATZ. The removal rate of ATZ by US/PMS was 45.85%. When compared with US alone, the degradation efficiency of ATZ was 32.43% higher, mainly because US can stimulate PMS to generate SO_4_^−^• and HO•, and SO_4_^−^•- and HO• have better degradation effects on ATZ [25,30]. Research by Dionysiou [31] indicated that the reaction rate of TBA to HO• and SO_4_^−^• is 3.8–7.6 × 10^8^ M^−1^·s^−1^ and 4–9.1 × 10^5^ M^−1^·s^−1^, respectively. The experiment conducted by Buxton [32] showed that the reaction rate of ETA to HO• and SO_4_^−^• is 1.2–2.8 × 10^9^ M^−1^·s^−1^ and 1.6–7.7 × 10^7^ M^−1^·s^−1^. Therefore, when HO• and SO_4_^−^• coexist in the reaction system, HO• can be captured by TBA, and HO• and SO_4_^−^• by ETA. From Figure 6b–d, both TBA and ETA can effectively inhibit the degradation of ATZ by US/PMS, and ETA has a stronger inhibitory effect than TBA. HO• and SO_4_^−^• exist in the US system simultaneously. The degradation rate of US to ATZ decreased from 45.85% to 16.36% and 8.28%, respectively, when 48 mg/L TBA and 48 mg/L ETA were maintained in PB at a pH of 7. Therefore, the degradation of ATZ by US/PMS at a pH of 7 was dominated by free radical oxidation.

### 3.7. Effect of Common Anion Concentration in Water on PMS Degradation of ATZ

The influence of common anions in waters as Cl^−^, HCO_3_^−^, and NO_3_^−^ on the degradation of ATZ by UV/PMS in PB of 1.25 mmol/L was shown in Figure 7 when the pH of PB, US intensity, PMS concentration, temperature, and ATZ concentration were 7, 0.88W/mL, 200 μmol/L, 20 °C, and 1.25 μmol/L, respectively. According to Figure 7, Cl^−^, HCO_3_^−^, and NO_3_^−^ with the same concentration showed an inhibitory effect on ATZ degradation by the US/PMS system. It is easy to observe that the inhibition ability of the three kinds of ions, from largest to smallest, is in the order of Cl^−^, HCO_3_^−^, and NO_3_^−^. Specifically, the degradation efficiency of the three kinds of ions on ATZ is reduced from 45.85% to 23.72%, 34.93%, and 46.75%, respectively, after adding 0.1 mmol/L Cl^−^, HCO_3_^−^, and NO_3_^−^ into the US/PMS system. The reaction rates of Cl•, Cl_2_^−^•, ClOH^−^• and CO_3_^−^• with ATZ were lower than those of HO• and SO_4_^−^• with ATZ. The inhibitory effect of Cl^−^ was slightly stronger than that of HCO_3_^−^ mainly because the secondary reaction constant between Cl_2_^−^• and ATZ was lower than that between CO_3_^−^• and ATZ. The weak inhibitory effect of the compound is mainly caused by the following two reasons: first, NO_3_^−^ can react with SO_4_^−^• to form NO_3_• with a high redox potential, and NO_3_• can also participate in the degradation of the ATZ reaction. Second, objectively speaking, the concentration of NO_3_^−^ is much higher than that of ATZ in the US/PMS system [33,34,35]. While the reaction rate of NO_3_^−^ and SO_4_^−^• is very low, a large number of NO_3_^−^ and SO_4_^−^• still compete with ATZ in the US/PMS system, which makes NO_3_^−^ show weak inhibition on the macro level (the main equations are shown in Equations (4) to (15)) [23,26,27,36,37,38,39,40,41,42,43].
HO•+HCO_3_^−^→CO_3_^−^• +H_2_O *K* = 8.60 × 10^6^ M^−1^·s^−1^(4)
SO_4_^−^•+HCO_3_^−^→CO_3_^−^• +HSO_4_^−^*K* = 2.80 × 10^6^ M^−1^·s^−1^(5)
CO_3_^−^•+ATZ→products *K* = 6.20 × 10^6^ M^−1^·s^−1^(6)
HO•+Cl^−^→ClOH^−^• *K* = 4.30 × 10^9^ M^−1^·s^−1^(7)
ClOH^−^•+Cl^−^→Cl_2_^−^•+OH^−^*K* = 1.0 × 10^5^ M^−1^·s^−1^(8)
SO_4_^−^•+Cl^−^→Cl• +SO_4_^2-^*K* = 3.0 × 10^9^ M^−1^·s^−1^(9)
Cl•+Cl^−^→Cl_2_^−^• *K* = 8.50 × 10^9^ M^−1^·s^−1^(10)
Cl_2_^−^•+ATZ→products *K* = 5.0 × 10^4^ M^−1^·s^−1^(11)
SO_4_^−^•+NO_3_^−^→NO_3_• +SO_4_^2-^*K* = 5.0 × 10^4^ M^−1^·s^−1^(12)
NO_3_•+ATZ→products(13)
SO_4_^−^•+ATZ→products *K* = 2.59 × 10^9^ M^−1^·s^−1^(14)
HO•+ATZ→products *K* = 3.0 × 10^9^ M^−1^·s^−1^(15)

### 3.8. Kinetic Analysis of ATZ Degradation by US/PMS

According to the research of Simonin [44], the kinetic model of ATZ degradation by O_3_ was established according to the following kinetic equation, and the quasi first order kinetic equation is as follows:
Ln(*C/C*_0_)=-*K*_1_*t*(16)

*C*—ATZ concentration at any time, μmol/L;

*C*_0_—ATZ concentration at time 0, μmol/L;

*K*_1_—quasi first order rate constant, min^−1^.

In PB of 1.25 mmol/L at a pH of 7, when the concentration of Cl^−^, HCO_3_^−^, NO_3_^−^, and ETA was 1.25 μmol/L, the quasi first order reaction kinetics of ATZ degradation by US/PMS was fitted using Ln(*C*/*C*_0_)(y) as the Y-axis and t(x) as the X-axis when the US intensity, PMS concentration, temperature, and ATZ concentration were respectively 0.88 W/mL, 200 μmol/L, 20 °C, and 1.25 μmol/L. Its dynamic fitting curves were shown in Figure 8 and the parameters for the quasi first order kinetic fitting equation were shown in Table 2. According to Figure 8 and Table 2, the ATZ degradation kinetics by US/PMS at different PMS concentrations were consistent with the quasi first order reaction kinetics, and ETA had the strongest inhibitory effect on the ATZ degradation of US/PMS under the same concentration. ETA reduced the degradation rate of US/PMS to ATZ by 10.91%, and the inhibitory effect of Cl^−^, HCO_3_^−^, and NO_3_^−^ on ATZ degradation by US/PMS was not significantly different under the concentration of 1 mmol/L. The mechanism of the three ion inhibitory effects is described in detail above and will not be described in this paper. Research results by Li and others [45] discovered that the degradation of 1,1,1-trichloroethane and 1,4-dioxane by US/PS were consistent with the first order reaction kinetics, which is similar to the results of this experiment.

### 3.9. Analysis of ATZ Degradation Products and Degradation Pathway by US/PMS

The degradation products of ATZ by US/PMS were analyzed by HPLC-ESI-MS (cationic mode), and the degradation path was speculated. The three samples were extracted for 5, 20, and 60 min in the process of the experiment. Then the first-order mass spectrum scanning was conducted, and the total ions and extracted ions were analyzed. According to the mass spectrum, total ion chromatogram and extracted ion flow diagrams from the three samples extracted at 5, 20, and 60 minutes. The mass-to-charge ratio of the main degradation products of ATZ was 128, 129, 146, 156, 172, 174, 188, 198, 214, 218, and 232. The relative molecular weight of ATZ is 216, m/z218 has a molecular weight 2 greater than ATZ. Thus m/z218 is produced when the methyl group is replaced by the hydroxyl group in the process of ATZ degradation, that is 2-chloro-4-diethylamino-6-hydroxyisopropyl atrazine, CDHA. m/z198 has a molecular weight 18 less than ATZ. Thus, in the process of ATZ degradation, Cl atoms are replaced by hydroxyl groups to produce 2-hydroxy-4-diethylamino-6-isopropylamino atrazine, HDIA. m/z156 has a molecular weight 42 less than HDIA, which is the molecular weight of isopropyl. Thus, m/z156 is regarded as the isopropyl product of HDIA, namely 2-hydroxy-4-diethylamino-6-amino atrazine, HDAA. m/z174 has a molecular weight 42 less than HDIA, which is the molecular weight of isopropyl. Thus, m/z174 is regarded as isopropyl ATZ, namely 2-chloro-4-diethylamino-6-amino atrazine, CDAA. m/z188 has a molecular weight 28 less than HDIA, which is the molecular weight of ethyl. Thus, m/z188 is regarded as desethyl ATZ, that is 2-chloro-4-amino-6-isopropylamino atrazine, CAIA. m/z146 has a molecular weight 28 less than HDIA, which is the molecular weight of ethyl. m/z146 has a molecular weight 42 less than CAIA, which is the molecular weight of isopropyl. Thus m/z146 is regarded as deethylated isopropyl ATZ, that is 2-chloro-4,6-diamino atrazine, CDA. m/z128 has a molecular weight 18 less than m/z146 and 28 less than m/z156. Thus m/z128 is regarded as chlorine ions of CDA replaced by hydroxyl groups or deethyl product of HDAA, that is 2-hydroxy-4,6-diamino atrazine, HDA. m/z129 has a molecular weight 1 greater than HDA. Therefore, m/z129 was produced by HAD amidogen, which was replaced by hydroxy, namely 2,4-dihydroxy-6-amino atrazine, DAA. m/z232 has a molecular weight 16 greater than ATZ, which is the molecular weight of hydroxy. Therefore, in the process of ATZ degradation, a hydrogen atom was replaced by a hydroxyl group to produce 2-Chloro-4-hydroxyethylamino-6-isopropyl atrazine, CHIA. m/z214 has a molecular weight 18 less than CHIA. Therefore, m/z214 may be the product of the CHIA chloride ion being replaced by a hydroxyl group, that is 2-hydroxy-4-hydroxyethylamino-6-isopropyl atrazine, HHIA. It can, thus, be seen that the degradation of ATZ by US/PMS was mainly conducted by dealkylation, dechloridation, and hydroxylation, which are similar to the research results of Ji [20], Javed [23], Wu [46], and others. The degradation path of ATZ is shown in Figure 9.

## 4. Conclusions

In this research, the US/PMS based degradation of ATZ in phosphate buffer is investigated. The degradation mechanism, oxidation kinetics, and degradation products of ATZ by US/PMS are analyzed. The influences of reaction temperature, PMS concentration, ATZ concentration, pH value, and US intensity on the degradation efficiency of ATZ are discussed in detail. The main results are presented as follows. The higher the temperature is, the higher the degradation efficiency of US/PMS to ATZ is. The temperature variation within the normal temperature range (10–25 °C) has a significant effect on the efficiency of ATZ degradation by US/PMS. Within the range of existing experimental settings, the greater the concentration of PMS is, the greater the intensity of US, the higher the pH value is, and the higher the degradation efficiency of US/PMS on ATZ is. Since the concentration of ATZ in the reaction system increases gradually, the efficiency of ATZ degradation by US/PMS decreases continuously. US can activate PMS. HO• and SO_4_^−^• coexist in the US/PMS system at the same time. The degradation of ATZ by US/PMS is dominated by free radical oxidation degradation. At a pH of 7, US alone accounted for 29.29% of the total ATZ removal rate. Cl^−^, HCO_3_^−^, and NO_3_^−^ showed inhibitory effect on the ATZ degradation by the US/PMS system under the concentration setting, and the inhibitory ability of the three ions was in the order of Cl^−^, HCO_3_^−^, and NO_3_^−^ from large to small. The reaction kinetics of ATZ degradation by US/PMS was consistent with Quasi first order reaction kinetics. The degradation of ATZ by US/PMS was mainly realized by dealkylation, dechloridation, and hydroxylation, but the triazine ring was not degraded. A total of 10 ATZ degradation intermediates were found by product analysis. The presented research has a high potential for the detoxification of water contaminated with atrazine.

## Figures and Tables

**Figure 1 ijerph-16-01781-f001:**
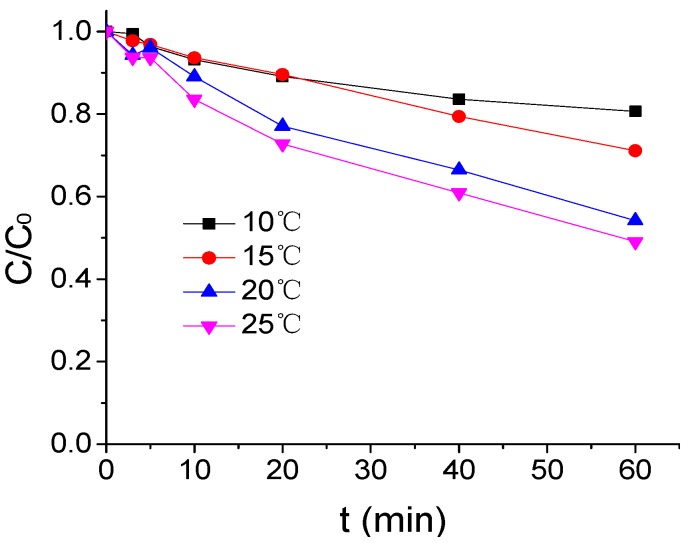
The ATZ removal rate under a different temperature (*C*_0_ = 1.25 μmol/L).

**Figure 2 ijerph-16-01781-f002:**
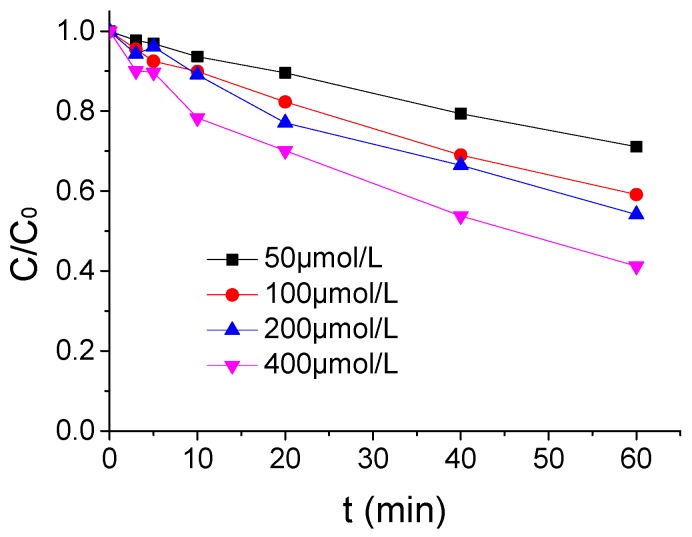
The ATZ removal rates under a different PMS density (*C*_0_ = 1.25 μmol/L).

**Figure 3 ijerph-16-01781-f003:**
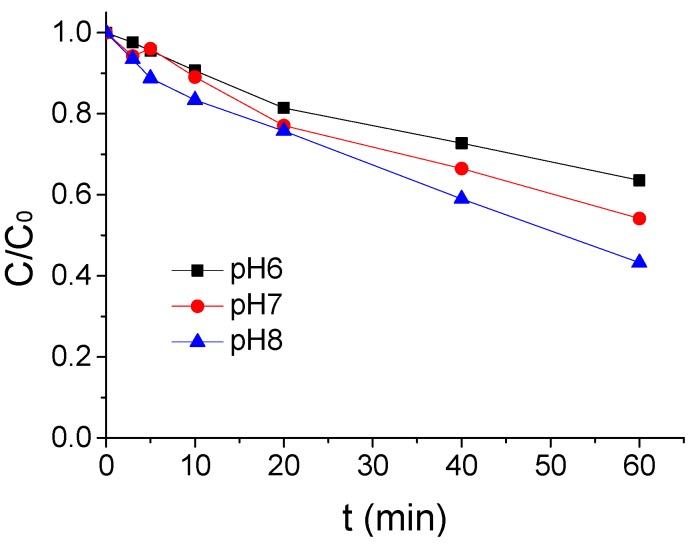
The ATZ removal rate under a different pH (*C*_0_ = 1.25 μmol/L).

**Figure 4 ijerph-16-01781-f004:**
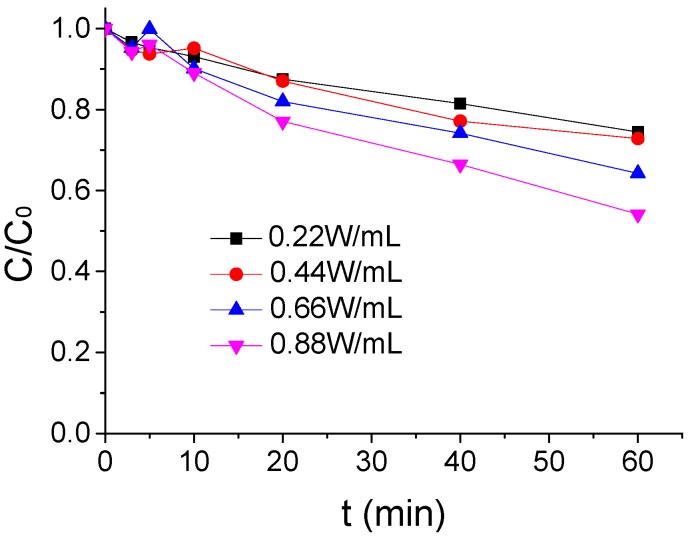
The ATZ removal rate under a different US intensity (*C*_0_ = 1.25 μmol/L).

**Figure 5 ijerph-16-01781-f005:**
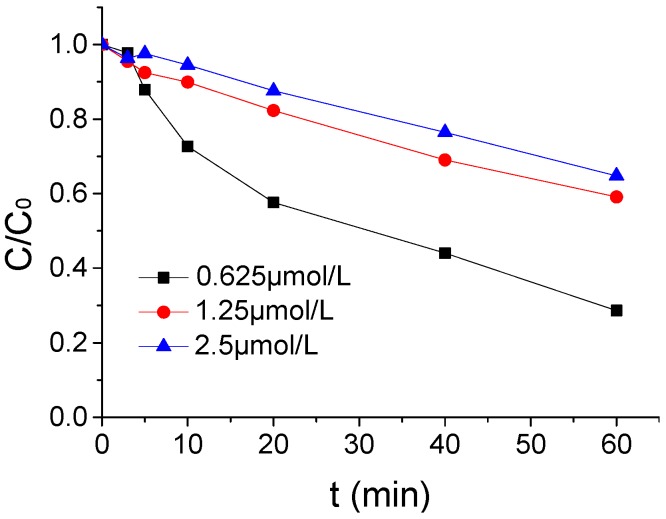
The ATZ removal rates under different ATZ density *C*_0._

**Figure 6 ijerph-16-01781-f006:**
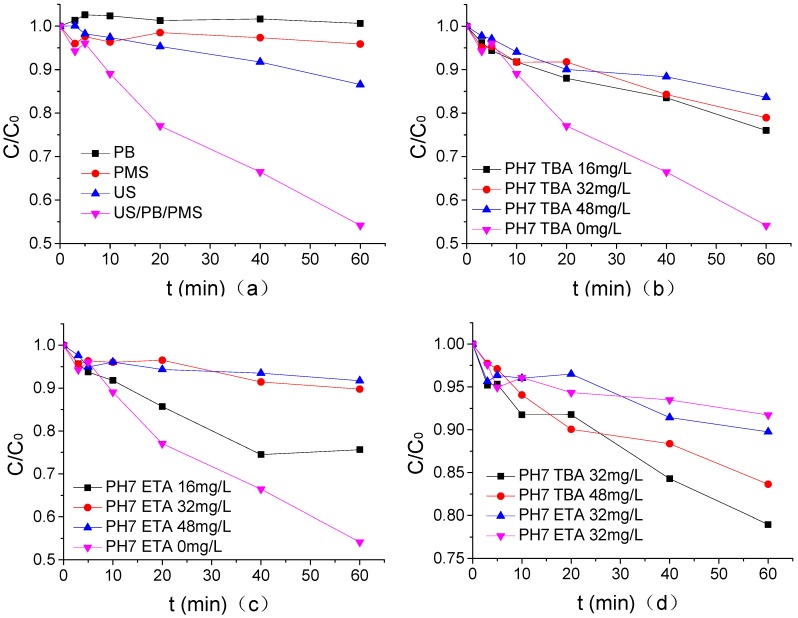
(**a**) The analysis of the oxidation effect of each component in the US/PMS system. (**b**) The effect of TBA on the degradation of ATZ by US/PMS in PB at pH 7. (**c)** The effect of ETA on the degradation of ATZ by US/PMS in PB at pH 7. (**d**) The Comparison of TBA and ETA on US/PMS degradation ATZ at a pH of 7. The initial concentration of *C*_0_ is 1.25 μmol/L.

**Figure 7 ijerph-16-01781-f007:**
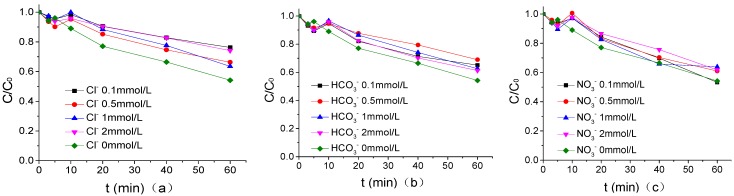
(**a**) The effect of Cl^−^ on the degradation of ATZ by US/PMS in PB at pH 7. (**b**) The effect of HCO_3_^−^ on the degradation of ATZ by US/PMS in PB at pH 7. (**c**) The effect of NO_3_^−^ on the degradation of ATZ by US/PMS in PB at pH 7. The initial concentration *C*_0_ = 1.25 μmol/L.

**Figure 8 ijerph-16-01781-f008:**
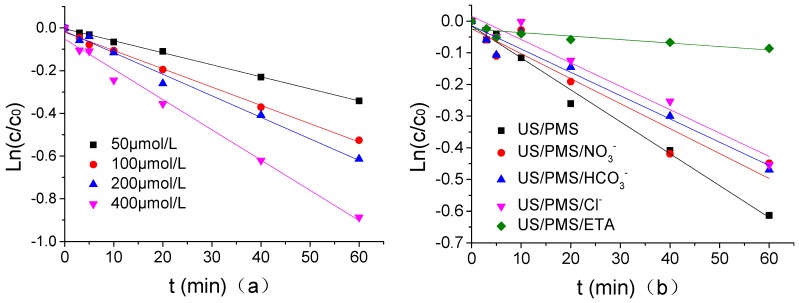
(**a**) The kinetics of quasi - first - order reaction of US/PMS degradation ATZ under different PMS density. (**b**) The kinetics of quasi - first - order reaction of US/PMS degradation ATZ under a different PMS density and different reaction systems. The initial concentration of *C*_0_ is 1.25 μmol/L.

**Figure 9 ijerph-16-01781-f009:**
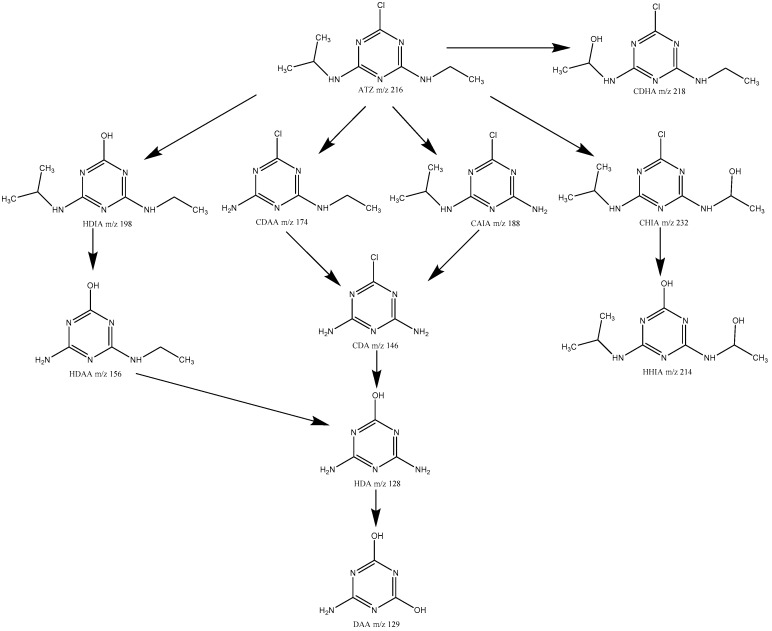
The possible degradation pathway of ATZ.

**Table 1 ijerph-16-01781-t001:** The preparation method of NaH_2_PO_4_-NaOH buffer.

pH	0.2 mol/L NaH_2_PO_4_ (mL)	0.2 mol/L NaOH (mL)
6	250	28.50
7	250	148.15
8	250	244.00

**Table 2 ijerph-16-01781-t002:** The kinetics parameters of US/PMS degradation ATZ.

Reaction System		*K*_1_ (min^−1^)	*R* ^2^
**US/PMS**	50 μmol/L	−0.00563	0.99875
	100 μmol/L	−0.00856	0.99571
	200 μmol/L	−0.01008	0.98788
	400 μmol/L	−0.01415	0.98768
US/PMS/NO_3_^−^		−0.00788	0.89850
US/PMS/HCO_3_^−^		−0.00735	0.94938
US/PMS/Cl^−^		−0.00738	0.95685
US/PMS/ETA		−0.00110	0.68884

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
