# Peer review of "Mechanism and Kinetic Analysis of Degradation of Atrazine by US/PMS"

_ijerph, 2019, doi:10.3390/ijerph16101781_

Round 1

Reviewer 1 Report

I n this article, the effect of US/PMS oxidative degradation to ATZ under different conditions was investigated in phosphate buffer solution, and its degradation mechanism, oxidation kinetics and degradation products were analyzed and discussed. The methodology and results are well presented, and the data and conclusions are reliable and convincing. However, revisions are needed prior to a possible publication in this journal.

1. The abstract should be rewritten. And there are too many grammatical errors and inconsistent typefaces in this manuscript which affect reading severely, the authors should edit this manuscript more carefully. English also should be polished.

2. The full name for abbreviations (such as ATZ, US, PMS, PB) should be presented at the first time.

3. Line 54-56, the removal methods of ATZ should be introduced in much more detail. And the following paper will be helpful in some aspects: Chemosphere, 2018, 200: 380-387; Applied Microbiology and Biotechnology, 2018, 102(17): 7597-7610; International Journal of Environmental Science and Technology, 2018, DOI: 10.1007/s13762-018-2148-2.

4. In this article, too many uncertain inferences are used by the authors, and these inferences cannot let readers convinced. The authors should do some experiments to testify those inferences or at least give some appropriate references. For example, Line 104-106, Line 156-158, Line 187, Line 219-222.

5. Line 108-112, the authors writed that the removal rate of ATZ increased more significantly compared with the temperature from 10 ℃ to 15℃ and from 20℃ to 25℃. However, why the removal rate of ATZ increased slowly when the temperature increased from 20 to 25 compared with 15 ℃ to 20℃.

6. Line 115, the 3.2 was wrong.

7. Line 122-124, why the removal rate of ATZ did not change much?

8. In the figures, the meaning of C and C0 should be illustrated.

9. The reaction time was 60 min in all figures, however, the degradation was not balanced at this time. Why not prolong the reacted time? And why didnot have the example about the effet of the reaction time?

10. Line 131, “From the figure in Fig.3” replaced by “From the Fig.3”.

11. Line 145-147, the reaction rate constants shoud be written in the equations.

12. Line 165, the sentence was wrong.

13. Table 2, it should be written by English.

14. The Conclusions parts should be rewritten.

15. Check the format in References parts.

Author Response

Dear Editor,

Thank you for reviewing our manuscript and giving us valuable suggestions to introverts quality. The authors have carefully studied the comments by the reviewers. We thank the reviewers for the thoughtful comments leading to the improvement of our manuscript. All the changes in the updated manuscript are highlighted in Red. The following is the response to the reviewers’ comments.

B. Responses to Reviewer #1

[Comment#1] The abstract should be rewritten. And there are too many grammatical errors and inconsistent typefaces in this manuscript which affect reading severely, the authors should edit this manuscript more carefully. English also should be polished.

[Response#1] The authors thank the reviewer for carefully reading the manuscript. We have rewritten the abstract. Please see page 1. Also, the authors have proofread the spelling and grammar of manuscript.

[Comment#2] The full name for abbreviations (such as ATZ, US, PMS, PB) should be presented at the first time.

[Response#2] The authors are thankful to the reviewer for the foregoing valuable comment. The authors have checked all the abbreviations used in this manuscript carefully. The full name of each abbreviation is presented.

[Comment#3] Line 54-56, the removal methods of ATZ should be introduced in much more detail. And the following paper will be helpful in some aspects: Chemosphere, 2018, 200: 380-387; Applied Microbiology and Biotechnology, 2018, 102(17): 7597-7610; International Journal of Environmental Science and Technology, 2018, DOI: 10.1007/s13762-018-2148-2.

[Response#3] The authors are thankful to the reviewer for the foregoing valuable comment.  In the revised manuscript, we have presented a more detailed introduction for removal methods of ATZ. Please see the first paragraph in page 2. Moreover, the recommended excellent works have been added into this paragraph.

[Comment#4] In this article, too many uncertain inferences are used by the authors, and these inferences cannot let readers convinced. The authors should do some experiments to testify those inferences or at least give some appropriate references. For example, Line 104-106, Line 156-158, Line 187, Line 219-222.

[Response#4] The authors are thankful to the reviewer for the foregoing valuable comment. In the revised manuscript, we have cited some appropriate references for our statement. Please see lines 113-116, lines 166-169, lines 195-196, and lines 226-229 of the revised manuscript.

[Comment#5] Line 108-112, the authors writed that the removal rate of ATZ increased more significantly compared with the temperature from 10 to 15 and from 20 to 25. However, why the removal rate of ATZ increased slowly when the temperature increased from 20  to 25compared with 15 to 20.

[Response#5] The authors are thankful to the reviewer for the foregoing valuable comment.  As the reaction temperature increases, the effect of temperature variation on the US/PMS based removal rate of ATZ will be less pronounced.  The same phenomenon was also observed by Su et al. [25]in the research of the degradation efficiency of the antibiotics amoxicillin in aqueous solution. They found that the activation of sulphate radicals under ultrasound irradiation present nonlinear correlation with temperature variation.

[Comment#6] Line 115, the 3.2 was wrong.

[Response#6] The authors thank the reviewer for carefully reading the manuscript. The title of section 3.2 is corrected. Please see line 129 in the revised manuscript.

[Comment#7] Line 122-124, why the removal rate of ATZ did not change much?

[Response#7] As the US intensity is constant the activation effect of  ultrasonic energy on the PMS is finite and maintaining stability. Thus, the variation of the PMS concentration does not change the removal rate of ATZ when US intensity is fixed.

[Comment#8] In the figures, the meaning of C and C0 should be illustrated.

[Response#8] The authors are thankful to the reviewer for the foregoing valuable comment. C represents ATZ concentration at any time and C0 is the concentration of ATZ at time 0. We have explained those two symbols in the revised manuscript, please see line 111-112 of the revised manuscript.

[Comment#9] The reaction time was 60 min in all figures, however, the degradation was not balanced at this time. Why not prolong the reacted time? And why didnot have the example about the effet of the reaction time?

[Response#9] The authors are thankful to the reviewer for the foregoing valuable comment. The reason why the reaction time was fixed at 60 min in those figures is given as follows. (1) The continuous working time of the ultrasonic equipment adopted in this experiment cannot exceed 60 min. (2) We desire the higher removal rate of ATZ in shorter reaction time, and longer reaction time may decrease cost-effectiveness. In future research, we may use better ultrasonic equipment and consider the effect of reaction time on the degradation efficiency.

[Comment#10] Line 131, “From the figure in Fig.3” replaced by “From the Fig.3”.

[Response#10] The authors thank the reviewer for carefully reading the manuscript. We have fixed the problem. Please see line 144 of the revised manuscript.

[Comment#11] Line 145-147, the reaction rate constants shoud be written in the equations.

[Response#11] The authors are thankful to the reviewer for the foregoing valuable comment. The reaction rate constants are added into those equations. Please see line 156-158.

[Comment#12] Line 165, the sentence was wrong.

[Response#12] Errors have been corrected. Please see line 175 in the revised manuscript.

[Comment#13] Table 2, it should be written by English.

[Response#13] Errors have been corrected. Please see the Table 2 in page 9 in the revised manuscript.

[Comment#14] The Conclusions parts should be rewritten.

[Response#14] The authors are thankful to the reviewer for the foregoing valuable comment. The authors have rewritten the section of conclusions. Please see page 11 in the revised manuscript.

[Comment#15] Check the format in References parts.

[Response#15] The authors thank the reviewer for carefully reading the manuscript. We have checked the formate of the references carefully.  In the revised manuscript,  all the references have been  reformed following the standard format of this journal.

Reviewer 2 Report

Mechanism and Kinetic Analysis of Degradation of 2 Atrazine by US/PMS
Yixin Lu, Wenlai Xu, Ying Zhang Na Deng and Jianqiang Zhang.

REVIEW

 This manuscript can be published after some revision. The authors have used correct chemical concentrations for the reactants, which is often neglected by other authors. The authors should state that this is a preliminary investigation that could be followed by a subsequent kinetics and mechanisms research project. They should also report that the reactions of atrazine with an oxidizing reagent has two reactants and therefore the reactants would follow second order kinetics. For this preliminary research, simple time studies are reported.
 For reactions that are kinetically second order, the chemical meaning of a pseudo first order reaction depends on which of the two reactants is in excess. For reactants a and b, the two possible pseudo first order kinetic rqte coefficients are ka = k/B and kb = k/A. The numerical values of ka and kb could be different by orders of magnitude. The authors should report which reactant was in excess.
 The captions of the figures should give the numerical values of the initial concentrations C0 .
 What is the difference between pseudo first order kinetics and quasi first order kinetics ?
 There is a practical question. Since it is necessary to not contaminate soil and water with the oxidizing agents or their reaction products, how would this laboratory study be used to control atrazine under field conditions ?

Author Response

Dear Editor,

Thank you for reviewing our manuscript and giving us valuable suggestions to introverts quality. The authors have carefully studied the comments by the reviewers. We thank the reviewers for the thoughtful comments leading to the improvement of our manuscript. All the changes in the updated manuscript are highlighted in Red. The following is the response to the reviewers’ comments.

Responses to Reviewer #2

[Comment#1] The authors should report which reactant was in excess.

[Response#1] The authors are thankful to the reviewer for the foregoing valuable comment. In the presented quasi first order reaction, the oxidant, namely the PMS, is the excess reactant. Also, we have added some details about this issue in section 3.8.

[Comment#2] The captions of the figures should give the numerical values of the initial concentrations C0.

[Response#2] The authors are thankful to the reviewer for the foregoing valuable comment. We have added the value of C0 in the captions of Fig.1 to Fig.8 except Fig.5. In the case of Fig 5, the value of C0 was already include in the picture.

[Comment#3] The captions of the figures should give the numerical values of the initial concentrations C0.

[Response#3] The pseudo first order kinetics is an equivalent, representation of quasi first order kinetics. In revised manuscript. In the revised manuscript, the pseudo first order kinetics will not be used anymore.

[Comment#4] There is a practical question. Since it is necessary to not contaminate soil and water with the oxidizing agents or their reaction products, how would this laboratory study be used to control atrazine under field conditions?

[Response#4] At present, there have been few reports on the degradation of ATZ by US/PMS. Therefore, in this paper, the effect of US/PMS oxidative degradation to ATZ under different conditions was investigated in phosphate buffer solution (PB), and its degradation mechanism, oxidation kinetics and degradation products were analyzed and discussed, which has certain reference value for chemical treatment of pesticide wastewater.

 In future research, we may optimize the PMS dosage to reduce the intermediate product.

The authors are thankful to the reviewer for carefully reading our manuscript and giving constructive suggestion.

Round 2

Reviewer 1 Report

All the comments and suggestions have been addressed carefully by authors. However, minor revision is required, before the manuscript is ready for publication.

1. Line 151-153, the reason about the removal rate of ATZ should be written in the manuscript.

2. Line 205, some words are lost about tht "C0".